# Feasibility of a conversation-based brief intervention in general practice to reduce post-traumatic symptoms after intensive care treatment—A qualitative analysis of the PICTURE study

Antina Beutel[1,2]*, Chris Maria Friemel[1], Daniela Lindemann[1], Linda Sanftenberg[1], Robert Philipp Kosilek[1], Konrad Schmidt[3,4], Cornelia Wäscher[3], Ulf-Dietrich Reips[2], Maggie Schauer[2], Tobias Dreischulte[1], Thomas Elbert[2], Jochen Gensichen[1], On behalf of the PICTURE study group¶

1 LMU University Hospital, Institute of General Practice and Family Medicine, Munich, Germany, 2 Department of Psychology, University of Konstanz, Konstanz, Germany, 3 Charité – Universitätsmedizin Berlin, Institute of General Practice and Family Medicine, Berlin, Germany, 4 Institute of General Practice, Faculty of Health Sciences Brandenburg, Brandenburg Medical School Theodor Fontane, Neuruppin, Germany

¶ Membership of PICTURE study group is provided in Supporting Information file S1 Acknowledgements.
* antina.beutel@uni-konstanz.de

## Abstract

There is currently a gap in the follow-up care of patients with post-traumatic stress disorder (PTSD) after intensive care. Long-term care is mainly provided within the setting of general practice. A conversation-based brief intervention carried out by general practitioners improved the symptoms of mild to moderate post-traumatic stress disorder sustainably. The aim of this study is to assess the subjective effectiveness and feasibility of short-term, primary care-based narrative exposure intervention from the perspective of participating general practitioners. Process-accompanying follow-up calls were made to all general practitioners in the intervention group to check the feasibility of the intervention and to reinforce the initial training. The survey was conducted by telephone between the second and third intervention sessions using an open-response questionnaire. The data analysis was carried out using the structuring qualitative content analysis according to Kuckartz. A total of N = 93 protocols from 87 general practitioners were analyzed. The intervention was assessed as useful and effective by most general practitioners. Nearly half of respondents considered permanent implementation in everyday practice, another third with certain limitations. The good learnability and practicability of the intervention as well as the patient's need were cited as beneficial factors. The greatest barriers to feasibility were seen in the high time expenditure and the lack of remuneration in standard care. Given the shortage of follow-up care for post-ICU PTSD symptoms and the substantial waiting times for trauma-focused psychotherapy, this low-threshold

**Data availability statement:** The analytical dataset, which includes deidentified patient data, and the code used to prepare and ana-lyse the data, is available in the research data repository of the Ludwig-Maximilians-University of Munich "Open Data LMU" and can be accessed at https://data.ub.uni-muenchen.de/557/. Access to the dataset is subject to our data use agreement, and further details can be found in the repository documentation. For enquiries about data use, potential collaborations or related projects, interested researchers are encouraged to contact the principal investigator of the study, Jochen Gensichen.

**Funding:** This study was supported by the German Research Foundation (DFG Grant: GE 2073/8-1). Our thanks also go to the German Center for Mental Health (DZPG), the Bavarian Research Practice Net-work (BayFoNet) and the Research Practice Network Berlin-BrandenburgThuringia (RESPoNsE). The funder had no role in the design and conduct of the study; collection, management, analysis and inter-pretation of the data; preparation and review or approval of the manuscript, or the decision to submit the manuscript for publication.

**Competing interests:** The authors have declared that no competing interests exist.

intervention delivered by general practitioners appears both feasible and well-positioned to help bridge a critical gap in care. Since the general practitioners surveyed emphasized the advantage of a quick and low-threshold treatment option for the care of patients with symptoms of PTSD after a stay on an intensive care unit, it seems sensible to ensure that general practitioners have access to training opportunities in the future. Additionally, diagnostics for Post-Intensive Care Syndrome should be implemented after intensive care.

## 1. Introduction

### 1.1. Long-term situation of former intensive care patients

Intensive care treatment can cause long-term sequelae. These are summarized in the symptom complex of Post Intensive Care Syndrome (PICS) as long-term physical, psychological and cognitive limitations that are temporally related to intensive care treatment [1]. Approximately 60% of survivors are affected [2,3]. In addition to the often severely impaired quality of life of those affected, loss of productivity due to absence from work and increased use of the healthcare system are also relevant consequences [4].

As an intensive care unit (ICU) stay is often accompanied by the experience of existential helplessness and fear, it is a potentially traumatizing life event and symptoms of post-traumatic stress disorder (PTSD) may occur. About 20% of survivors after intensive care treatment develop post-ICU PTSD [5]. As a result of the SARS-Cov2 pandemic, the prevalence is also expected to increase [6].

There are hardly any aftercare programs for those affected. However, functional limitations in everyday life and comorbidities already manifest themselves in sub-syndromal PTSD [7]. Even if the diagnosis has been made and a need for treatment has been identified, like in most other countries, in Germany there are usually long waiting times of three to nine months for a psychotherapy place [8]. For survivors of an ICU stay, the general practitioner´s office is the first and most important point of contact for follow-up care.

### 1.2. Brief primary care-based narrative exposure intervention

The PICTURE study ("PTSD after ICU survival") was developed to investigate the possibilities for follow-up care of former intensive care patients with post-traumatic symptoms. In the context of primary care, a brief talk therapy with case management is being tested, which is based on the proven Narrative Exposure Therapy (NET) [9,10], whose effectiveness is well documented [11]. As the perceptual, emotional and cognitive memories of traumatic events are often poorly integrated with their temporal and spatial context, NET employs targeted questioning to reconstruct the memory and anchor the event within the patient's autobiographical past. The short version of the NET was implemented for the first time in the PICTURE study [12].

The GPs received training in advance and were methodically supported in follow-up calls during the intervention [13,14].

PLOS Mental Health

### 1.3. Aims of this study

The study had two aims: (1) to identify factors that facilitate or impede the feasibility of the GP-delivered version of NET from the general practitioners' perspective, and (2) to assess their subjective evaluation of its effectiveness.

## 2. Methods

### 2.1. Ethics statement

The PICTURE study was approved by the ethics committee and is registered in the German Study Register under the number DRKS000 12589 [13]. This sub-study is included in it. The original approval document and all renewals are available from the publisher. All participants confirmed their consent in writing. This declaration of consent was also valid for the follow-up calls.

### 2.2. Study design: Follow-up calls

During the main PICTURE study, the majority of GPs in the intervention group were contacted by PICTURE team psychologists via telephone – typically between the second and third NET sessions – after scheduling the call in advance. The aim of each 25–40-minute telephone call was to consolidate methodological skills. To this end, key training content was revisited and tailored to the individual patient's stress profile. Additionally, any questions that had arisen during the first two NET sessions were clarified.

The telephone call was also used to collect subjective assessments regarding

1. feasibility,

2. the preparation provided by the training,

3. effectiveness,

4. the usefulness and transferability of the intervention to the GP context

(see S2 Text follow-up calls).
The research questions were:

1. Which facilitating and hindering factors can be identified for the feasibility of a GP version of NET from the GPs' perspective?

2. How do the GPs subjectively assess effectiveness?

The follow-up calls were made by a qualified psychologist (CF) and a master's student in psychology (AB) who was also an experienced nurse. There were no other people in the room during the calls. There were no personal relationships between the participants. The interviewees were informed about the professions of the callers and the aim of the survey during the study. They were also informed about the content of the calls when the appointments were made. The telephone calls were not recorded. However, a memory log was created in each case which was not subsequently presented to the interviewees for their information. There was no pilot phase and no further follow-up calls. All protocols were included in the data analysis, which ensured that the content was saturated.

This report adheres to the Consolidated Criteria for Reporting Qualitative Research (COREQ) [15] to ensure transparency and rigor in the presentation of qualitative findings (see S1 COREQ-Checklist in S3 Checklist).

### 2.3. PICTURE main study

The GP participants in the main PICTURE study were not selected but were asked to participate after their patients had been included in the study. Alternative GPs were recruited by the study team for those patients whose GPs had declined

to participate. The recruitment period for the main PICTURE study began on October 21, 2018, and ended on January 18, 2023. All participants confirmed their consent in writing. This declaration of consent was also valid for the follow-up calls.

The PICTURE study was approved by the ethics committee and is registered in the German Study Register under the number DRKS000 12589 [13]. This sub-study is included in it. The original approval document and all renewals are available from the publisher.

The results of the main PICTURE study were published in May 2025 [12].

## 2.4. Analysis

After completion of all follow-up calls, the first author conducted a structuring qualitative content analysis based on Kuckartz [16] using MAXQDA software (VERBI software). The template for the protocols was created by CF at the beginning of the PICTURE study and was not changed during the course of the study. Deductive and inductive approaches were combined to develop the category system. The deductive categories were based on the structure of the survey protocols. Their template was divided into five sections: 1. feasibility, 2. preparation provided by the training, 3. effectiveness, 4. usefulness/transferability, 5. comments. The categories were defined and provided with anchor quotations from the text material. A coding guide was created (see S4 Data for codebook). In the first coding round, inductive categories were defined from content-related text passages that could not be assigned to the deductive categories and also provided with anchor quotations. In repeated coding rounds, the level of abstraction was increased, and the category definitions were linguistically sharpened. Memos were created continuously during the coding rounds, and the entire analysis process was documented in a research logbook.

For quality assurance purposes, there was an ongoing collegial exchange with the PICTURE team at the Munich Study Center and in a qualitative research seminar at LMU Munich. To ensure inter-coder reliability, seven transcripts were coded independently by AB and CF. Those codes that showed insufficient agreement were discussed and the category system was revised again. All suggestions for better intersubjective comprehensibility of the category definitions were incorporated.

## 3. Results

A total of N = 93 follow-up calls were conducted in the period from July 2020 to May 2023, each lasting 25–40 minutes. Table 1 describes the demographic characteristics of the GPs in the intervention group (N = 160).

Of the N = 160 GPs in the intervention group, n = 116 carried out all three planned sessions, n = 6 had only two sessions, n = 3 had only one session and n = 35 had no session. In order to consider frequent questions asked by the GPs at the beginning of the recruitment phase and to give all participants the opportunity to strengthen methodological certainty, the follow-up calls examined here were implemented in the course of the study. These were conducted with n = 87 GPs between the second and third sessions by psychologists from the study team. N = 93 protocols were created (three GPs implemented two interventions, one GP carried out the intervention with four patients).

Three main aspects emerged to answer the research questions:

### 3.1. Assessment of the usefulness of the intervention

Two thirds of the GPs contacted emphasized that they considered the intervention to be useful overall. Three aspects were particularly emphasized:

- the brief primary care-based narrative exposure intervention (NET-PC) is a help in everyday life, it is a "very useful method that fulfills the purpose of short-term help by the GP" (PICTURE_30–428),

- NET-PC is easy to implement, it is an "intervention approach that is easy to learn, easy to use and uncomplicated" (PICTURE_31–696) and

**Table 1. Sample structure and characteristics of the intervention group.**

| Feature | Number (absolute/ %) |
|---|---|
| **Total intervention group:** | N = 160/100 |
| **Gender:** | |
| male | 81/51 |
| female | 79/49 |
| diverse | 0 |
| **Professional experience:** | |
| 0–3 years | 2/1,3 |
| 4–10 years | 12/7,6 |
| 11–20 years | 52/32,6 |
| 21–45 years | 73/46 |
| N/A | 21/13,1 |
| **Practice type:** | |
| Individual practice | 63/39 |
| Group practice | 68/42 |
| Medical care center | 12/8 |
| N/A | 17/11 |
| **Location of the practice:** | |
| Rural area | 13/8 |
| Rural municipality <5000 inhabitants | 14/9 |
| Small Town 5001–20,000 inh. | 38/24 |
| Medium-sized city 20,001–100,000 inh. | 26/16 |
| Large city >100,000 inh. | 65/41 |
| N/A | 4/2 |
| **Duration of doctor-patient relationship:** | |
| <1 year | 36/22,5 |
| 1–5 years | 49/30,6 |
| 6–15 years | 44/27,6 |
| 16–30 years | 19/11,9 |
| N/A | 12/7,5 |

This information is provided to enable a reader to understand from which population the sample was drawn.

- there is a need for it in everyday practice, the "topic is given far too little attention in practice" (PICTURE_31–893).

    More than two thirds of respondents were positive about participating in the study. This was "personally a great enrichment" (PICTURE_31–780). Almost half of the respondents particularly emphasized the aspect of gaining knowledge and raising awareness of the topic:

- The "therapy approach is useful, it has expanded one's own therapeutic 'toolbox'" (PICTURE_31–948),

- "Participation in the study is a benefit, as further knowledge of interviewing was acquired, which could also be transferred to everyday patient interviews" (PICTURE_31–961) and

- the "diagnostic view for PTSD was sharpened, the knowledge is valuable for the further professional career" (PICTURE_31–981).

Few GPs were critical of the usefulness of the intervention in the GP context. On the one hand, structural reasons were cited: The doctor "sees it critically that psychiatric care gaps should be covered by GPs; rather more therapist positions should be created" (PICTURE_31–773).

On the other hand, there were substantive reasons that questioned the usefulness of the intervention. Here, the participants mentioned their own excessive emotional stress and the mechanism of transference in the therapeutic conversation: "He himself would reach his limits with the topic of PTSD" (PICTURE_31–535), "It was difficult to keep a professional distance, the topic was a burden for her" (PICTURE_31–996), "Doctor went along emotionally, also felt anger" (PICTURE_31–990). A few participants expressed doubts about their own psychotherapeutic competence: "Doubts as to whether she is the right person professionally" (PICTURE_31–996).

Table 2 provides an overview of the arguments mentioned:

### 3.2. Subjective assessment of the effectiveness of the intervention

The statements on effectiveness must be viewed from two perspectives: They are not based on validated measurements, but on the subjective impressions of the GPs, which are purely symptom oriented. In addition, the follow-up calls took place between the second and third sessions, when the intervention had not yet been completed.

The fact that the intervention enabled "rapid stress reduction with simple means" was cited as beneficial for its effectiveness (PICTURE_31–966). In addition, the "psychoeducation alone helps to identify and minimize psychosomatic complaints" (PICTURE_30–528). Well over half of all respondents reported a high level of patient acceptance of the intervention. After just two sessions, the patients were "less overwhelmed by memories and felt less stressed" (PICTURE_50–310). The "patient is no longer afraid of the ICU, can now look at photos from the ICU stay again, [shows] less avoidance behavior" (PICTURE_31–815). Particular emphasis was placed on the relief provided simply by verbalizing the experience: "Patient was happy to be able to talk to a professional about the ICU; this made it possible to classify medically necessary measures and provided relief" (PICTURE_31–922), "Patient said that it was good for him to talk about his ICU memories, as he did not want to do this with his family and friends" (PICTURE_31–842). Psychological stabilization often seems to be linked to the somatic recovery process: "Patient is happy to have survived, has recovered well physically and mentally" (PICTURE_50–375).

The "clear strengthening of the doctor-patient relationship" was emphasized several times as beneficial (PICTURE_30–428).

The following were mentioned as barriers to effectiveness:

- Some interviewees expressed doubts about the inclusion diagnosis, "as the patient felt little distress due to ICU stay" (PICTURE_31–905).

**Table 2. Arguments for and against the usefulness of the intervention.**

| Usefulness given | Usefulness not given | |
|---|---|---|
| **NET-PC is a help in everyday life.** | Structural: | Content: |
| **NET-PC is easy to use.** | NET is not the task of GPs. | Your own psychological stress is too high. |
| **There is a need for this in everyday practice.** | | Doubts about your own psychotherapeutic competence. |
| **Gaining knowledge and raising awareness of the topic.** | | Mechanisms of transference in therapeutic conversation. |

**Bold:** Statements that *were mentioned frequently or very frequently, i.e., by more than 50% of respondents.*

- The presence of psychiatric comorbidities made it difficult to assess the effectiveness: "dominant depression overlaps PTSD symptoms, no clear improvement noticeable" (PICTURE_30–746). In this sense, it was also expressed that this brief primary care-based narrative exposure intervention was not the right therapeutic approach: "Talked at length about the difficulty of the particular (...) life issue of loneliness; NET is not the right therapeutic approach here" (PICTURE_31–528). "He believes in the effectiveness of brief intervention for monotrauma, but the patient's polytrauma requires more complex psychotherapy" (PICTURE_31–315).

- The low number of sessions was criticized: "She has doubts that three sessions are enough for her patient" (PICTURE_31–717).

- Almost half of the respondents commented on the persistence of PTSD symptoms: "Patient continues to have night-mares and sleep problems, is jumpy" (PICTURE_31–528), "Patient is very sensitive to acoustic triggers (e.g., siren), finds this stressful" (PIC-TURE_50–364).

- Persistent health fears on the part of the patient overshadowed the potential effectiveness of the intervention: "Patient still has fears of complications, especially bleeding to death" (PICTURE_31–958).

Few GPs commented on alleged side effects: "Patient has experienced a high level of stress since ICU, before that traumatic memories were painstakingly suppressed, this is no longer possible; the negative is catching up with him again" (PICTURE_31–519). "Patient reported that the lifeline still bothered him afterwards, as topics were discussed that had never been discussed before; his wife also knew nothing about it, which made him feel guilty" (PICTURE_31–797).

Table 3 provides an overview of the arguments mentioned:

### 3.3. Assessment of feasibility

The following arguments were given in response to the question of whether GPs could envisage NET-PC becoming a permanent fixture in everyday practice:

Nearly half of respondents would implement NET-PC in patient care without restriction in the future: "He has other patients he wants to treat, including traumatized refugees" (PICTURE_50–284). The ease of learning was mentioned positively: "He considers the practicability in everyday practice to be sensible, feasible and desir-able" (PICTURE_30–893). Approval of the implementation was reinforced by the positive experience that initial fears had not materialized: "Overall, the sessions did not cause any difficulties - contrary to what was feared" (PIC-TURE_31–853).

Table 3. Arguments for and against the subjective effectiveness of the intervention.

| beneficial | obstructive |
| --- | --- |
| **Be allowed to speak with a professional expert** | **Doubts about the inclusion diagnostics** |
| **High patient acceptance** | **Persistence of PTSD symptoms** |
| Low-threshold therapy offer | Insufficient number of sessions |
| If the somatic recovery process progresses at the same time | Psychiatric comorbidity (NET-PC is not the right therapeutic approach) |
| Rapid load reduction | Persistent patients´ health anxieties |
| Strengthening the doctor-patient relationship | Possible side effects |

**Bold:** Statements that were mentioned frequently or very frequently, i.e., by more than 50% of respondents.

Just less than a third of respondents reported that they could imagine implementing the intervention in everyday practice with restrictions, namely only for selected patients: "She would use it with suitable patients, but rather in individual cases" (PICTURE_30–061). "Feasibility yes, but only conceivable with former ICU patients with post-ICU PTSD, not in the context of other traumas, she wouldn't trust herself to do that" (PICTURE_31–858).

A few interviewees expressed a fundamental resistance to implement the intervention: "Therapy like this belongs in the hands of professional therapists" (PICTURE_31–528). Doubts about their own psychotherapeutic competence were mentioned here: "He tends to hand over trauma patients to specialists and doesn't feel competent enough himself" (PICTURE_31–773).

Two thirds of respondents cited structural barriers to the possible implementation of NET-PC in everyday practice. These were, in particular, the time required and the lack of remuneration:

- "Only possible with restrictions, if at all, as time demands are too high" (PIC-TURE_31–389).

- "Would always be an 'on top' offer in everyday practice, for which time would have to be freed up" (PICTURE_31–694).

- "The time frame is often difficult to realize, i.e., the sessions should always be after the practice closes" (PICTURE_31–741).

- "Financial remuneration is not appropriate for basic psychosomatic care" (PICTURE_31–832).

As the entire period of the SARS-Cov2 pandemic coincided with the recruitment phase of the PICTURE study, the GPs who were particularly challenged during this phase also commented on aspects of feasibility during the pandemic: "Implementation in everyday practice is conceivable, but only after the pandemic due to time constraints (PICTURE_31-471). "She would like to work intensively with her patients in this way much more often; this is currently not possible due to the pandemic" (PICTURE_31–717). In the event of continuation, it would have to be clarified how post-ICU PTSD could be diagnosed, as this was carried out by the study team during recruitment: "If implemented, the question of PTSD diagnosis is unclear" (PICTURE_31–825).

The need for regular implementation was also mentioned: "Frequent implementation is necessary in order to have one's own security and to be able to conduct the sessions well" (PICTURE_30–893).

Table 4 provides an overview of the arguments mentioned:

**Table 4. Arguments for and against the feasibility of the intervention.**

| unreservedly „Yes " | „Yes"with restrictions | basically „No " |
|---|---|---|
| **Other patients in the practice who have a need** | **Only for selected patients** | PTSD treatment is not the task of general practitioners. |
| **Easy to implement in everyday practice** | **Structural and individual barriers** | |
| **Intervention is easy to learn** | **Time demands too high** | |
| **Initial fears did not materialize.** | **Financial remuneration not appropriate** | |
| | Doubts about your own psychotherapeutic competence | |
| | Diagnostics unclear in the case of continuation | |
| | Frequent implementation is necessary for methodological safety | |

**Bold:** Statements that were mentioned frequently or very frequently, i.e., by more than 50% of respondents.

## 4. Discussion

### 4.1. Summary

This qualitative study examined general practitioners´ subjective assessment of the NET-PC brief intervention regarding its usefulness, effectiveness and feasibility.

The majority of participating GPs rated the intervention positively, though a few questioned its utility. Most respondents observed rapid symptom reduction and high patient acceptance, which they viewed as evidence of effectiveness. However, several barriers emerged: concerns about diagnostic inclusion criteria, psychiatric comorbidities, patients´ health anxieties and potential aggravation of PTSD symptoms. Regarding future implementation, opinions were divided into three groups: Nearly half of respondents expressed willingness to implement NET-PC in routine practice without restriction. Another third would offer NET-PC only to selected individual patients, and the remainder opposed its use entirely. The high time demands and the inadequate financial remuneration for basic psychosomatic care were frequently cited as implementation barriers.

### 4.2. Classification of the results

GP´s who expressed concerns about the usefulness of NET-PC cited two main barriers: the emergence of transference dynamics during the therapeutic conversation and doubts about their own psychotherapeutic competence. These concerns warrant serious consideration as not all GPs are capable of trauma therapy - particularly those with unresolved psychological trauma of their own.

With regard to effectiveness, the doubts expressed about the inclusion diagnostics, namely that the patients were not traumatized at all, must be placed in the overall context insofar as only mild to moderately traumatized patients were included. The boundaries to acute stress reactions are fluid here, so that many patients may have experienced a reduction in symptoms over time in the sense of self-management. A presumed intensification of symptoms through intervention should be understood against the background that the verbalization and contextualization of the experience can initially increase psychological arousal. However, this is seen as beneficial for the healing process: "Many patients only recognize what was actually threatening about the event for themselves through the slow and precise description of the traumatic event and recognize how the meanings learned in the trauma still determine their lives today" [17]. Some interviewees reported that they continued to observe PTSD symptoms in their patients. As not all sessions were completed at the time of the survey, a clear reduction in symptoms cannot yet be expected.

The limiting factor of time is possibly put into perspective against the backdrop of the pandemic, which completely coincided with the study period during which the general practice teams in particular had to cope with a heavy workload. This makes the high overall approval rates all the more remarkable.

The frequently cited criticism of inadequate remuneration is based on the standardized evaluation standards for psychosomatic primary care in Germany. For example, a GP with the appropriate qualifications can charge a total of around €72 (equivalent to £61 or $80) for three 45-minute psychosomatic consultations per quarter, depending on the regional point value and individual billing modalities.

### 4.3. Implication for research and clinical practice

PTSD is widespread in primary care [18]. Initial diagnosis and low-threshold treatment for mild to moderately stressed patients should therefore take place in this setting. Due to the frequent avoidance behavior despite impaired everyday functionality and quality of life, patients with PTSD [19] rarely seek support from psychotherapists. Due to the usually long-standing doctor-patient relationship in general practice, the GP practice can be a low-threshold point of contact. There are no long waiting times for psychotherapy appointments and the experience of stress is quickly reduced [12]. As a result, those affected can get on with their daily lives again sooner, which is also of great importance in terms of health

economics. This needs-adapted approach has already proven itself in other areas of psychiatric healthcare in the stepped care model [20,21].

In the main PICTURE study, the NET intervention was highly time-limited consisting of only three sessions. This brief and concise treatment constrained the potential for substantial stress reduction, especially for patients with more severe post-traumatic stress. In these cases, the involvement of para-medical staff such as medical assistants could allow more time to deal with trauma. Previous studies have shown that this can be a possible solution [22,23], and Kaiser et al. [24] have recently published a manual for such a treatment.

It therefore seems sensible to ensure that GPs have access to training opportunities in the future. In addition, diagnostics for post-traumatic symptoms after intensive care should be implemented in follow-up settings.

The evaluation of the brief primary care-based narrative exposure intervention from the perspective of the treated patients has already been published elsewhere [25].

### 4.4. Strengths and weaknesses of our study

One strength of the present observation is that a selection bias is rather low. This aspect is due to the recruitment method of the main study, in which the participating GPs were not explicitly selected, but were randomized into the intervention group of the main study via the consent of their patients. As a result, the GPs took part in the follow-up calls examined here without further selection.

A weakness of this sub-study is that the effectiveness could only be assessed from the subjective perspective of the respondents and only during the course of the intervention. Furthermore, socially desirable responses towards the study participants can never be ruled out.

As the main purpose of the follow-up calls was to increase methodological adherence, a number of respondents did not comment on all aspects.

### 5. Conclusion

Even before the SARS-Cov2 pandemic, the prevalence of post-ICU PTSD was conservatively estimated at around 20% of ICU survivors [5]. Post-pandemic, an increase can be assumed [6], not least against the background of the demographically-related increase in multimorbidity of the patients treated. With regard to the limited everyday functionality of those affected and the avoidance of chronification of symptoms, concepts for reducing the long-term consequences of intensive care treatment should be expanded. As the therapy offered by NET-PC is low-threshold for patients and GPs alike [14], integration into standard care should be considered. In view of the high trauma follow-up costs, this would also be advantageous in terms of health economics [4]. The procedure has proven to be feasible and often successful [12]. A slightly higher intensity with support from the GP team could lead to even better results. Especially as psychotherapists often do not have medical expertise about the ICU setting and the possibility of factually classifying what is experienced there.

### Supporting information

**S1 File. Acknowledgements.**
(PDF)

**S2 Text. Template follow-up calls.**
(PDF)

**S3 Checklist. COREQ Checklist.**
(PDF)

**S4 Data. Codebook.**
(PDF)

## Author contributions

**Conceptualization:** Antina Beutel, Linda Sanftenberg, Robert Philipp Kosilek, Konrad Schmidt, Ulf-Dietrich Reips, Maggie Schauer, Tobias Dreischulte, Thomas Elbert, Jochen Gensichen.

**Data curation:** Daniela Lindemann, Linda Sanftenberg, Robert Philipp Kosilek, Konrad Schmidt.

**Formal analysis:** Antina Beutel.

**Funding acquisition:** Konrad Schmidt, Jochen Gensichen.

**Methodology:** Antina Beutel, Chris Maria Friemel, Ulf-Dietrich Reips, Maggie Schauer.

**Project administration:** Daniela Lindemann, Robert Philipp Kosilek, Cornelia Wäscher, Jochen Gensichen.

**Supervision:** Chris Maria Friemel, Tobias Dreischulte, Thomas Elbert, Jochen Gensichen.

**Validation:** Chris Maria Friemel.

**Writing – original draft:** Antina Beutel.

**Writing – review & editing:** Chris Maria Friemel, Linda Sanftenberg, Maggie Schauer.

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
