## [Decision Letter · Decision Letter 0]

4 Nov 2025

PMEN-D-25-00399

Feasibility of a conversation-based brief intervention in general practice to reduce post-traumatic symptoms after intensive care treatment - a qualitative analysis of the PICTURE study.

PLOS Mental Health

Dear Dr. Beutel,

Thank you for submitting your manuscript to PLOS Mental Health and I am sorry for the delay. After careful consideration of the reviewer reports, we feel that it has merit but does not fully meet PLOS Mental Health’s publication criteria as it currently stands. Therefore, we invite you to submit a revised version of the manuscript that addresses the points raised during the review process.

Please address all of the comments raised, which you can find at the end of this report and attached. We also request that you ensure that the limitations of the study are clear in light of the comments from reviewer 2.

We look forward to receiving your revised manuscript.

Kind regards,

Dr Karli Montague-Cardoso

Staff Editor

PLOS Mental Health

Journal Requirements:

1. Please send a completed 'Competing Interests' statement, including any COIs declared by your co-authors. If you have no competing interests to declare, please state "The authors have declared that no competing interests exist".

Reviewers' comments:

Reviewer's Responses to Questions

**Comments to the Author**

1. Does this manuscript meet PLOS Mental Health’s publication criteria?

Reviewer #1: Yes

Reviewer #2: Partly

2. Has the statistical analysis been performed appropriately and rigorously?

Reviewer #1: Yes

Reviewer #2: No

3. Have the authors made all data underlying the findings in their manuscript fully available (please refer to the Data Availability Statement at the start of the manuscript PDF file)?

Reviewer #1: Yes

Reviewer #2: Yes

4. Is the manuscript presented in an intelligible fashion and written in standard English?

Reviewer #1: Yes

Reviewer #2: No

Reviewer #1: PLOS - trauma-focused narrative exposure therapy Germany

Introduction

The first part of the introduction is actually good, allowing the statement of the importance of the intervention under study, nevertheless, the flow in the second part is lost as there is a lack of:

Definition of what is trauma-focused narrative exposure therapy is and a brief literature review on its uses and effectiveness. I could not follow the argument until I looked in the literature for the NET and Picture study.

The following information would have been interesting and useful for the reader that is not familiar with this work, like the paragraph in the following article: Sanftenberg, L., Beutel, A., Friemel, C.M. et al. Barriers and opportunities for implementation of a brief psychological intervention for post-ICU mental distress in the primary care setting – results from a qualitative sub-study of the PICTURE trial. BMC Prim. Care 24, 113 (2023). https://doi.org/10.1186/s12875-023-02046-0

(Therefore, the PICTURE trial (“PTSD after ICU survival”) has been initiated to improve the follow-up care of ICU patients in primary care by testing a brief talking therapy with case management. The intervention is based on narrative exposure therapy (NET), a well-established combination of testimony therapy and cognitive-behavioral therapy, originally developed for low-income countries [17]. Assuming a disturbed memory formation for a traumatic event the NET aims to rebuild memory by a detailed and chronological review of the traumatic scenario in which fragmented trauma memories can be embedded. By using targeted questioning the therapist facilitates a vivid narration and mental re-exposure to the traumatic event in the patient.

Please include a review about the existing experience in the application of this tool and benefits according to the literature.

Additionally, the literature should describe potential factors that may influence the implementation and effectiveness of this program, allowing for later evaluation of the results in the discussion.

Methodology: well described, comprehensive, and includes the elements needed for qualitative study methodology reporting.

However, the study does not say where the questions came from or how the interview questions were created. How was it decided?

Correct the word” seven tra7nscripts” line 150.

Results

Categories that have 0 frequencies do not need to be reported.

For reflection: what is the point of integrating sociodemographic variables when you do not incorporate them in the explanation of the results? For example, were there differences in the acceptance of the opinion about feasibility according to years of experience, for example???

Discussion: what about comparing your results with similar studies made under similar constructs and conditions, more in-depth, in a more reflective way, not only saying it is similar or not to other studies, but also how it is similar and different and what it means for the German context?

It would be interesting to have a chart that explains the main codes generated and themes based on codes in order to follow how those who have analyzed the data have come up with the codes.

Reviewer #2: Best authors.

Thank you very much for submitting your manuscript to PLOS Mental Helath and that I was allowed to review it. Its theme is highly interesting, as many ICU survivors do suffer from longterm consequences that form PICS, comprising mental helath symptoms of PTSS and in some, also developing into PTSD. It is true that there are numerous ICU survivors who because of their post-ICU state, needs more help, and GPs are one of many groups of professionals who meet and treat these patients. GPs expressions are absolutely interesting to know more about. There is a need for more to be done and I salute your interest in the field and work done yet. Keep going on.

However, I consider that there are some major issues with the manuscript that needs to be handled before it can be re-considered. To my mind, it needs additional rounds of work by the whole group of authors, so that the rationale behind this qualitative sub-study are easier to follow, how the results relates to eachother, the meaning of the results in each cathegory and why you choose to present them and the the citations used to examplify each cathegory, also to better discuss and present the clinical impact of your findings. As post-ICU procedures may be very different in Germany than other countries such as Australia, England or others, a suggestion is if your manuscipt would benefit to be submitted to a German native paper?

I wish you all good.

**Do you want your identity to be public for this peer review?** For information about this choice, including consent withdrawal, please see our Privacy Policy

Reviewer #1: No

Reviewer #2: No

---

## [Editor Report · Decision Letter 1]

18 Dec 2025

Feasibility of a conversation-based brief intervention in general practice to reduce post-traumatic symptoms after intensive care treatment - a qualitative analysis of the PICTURE study.

PMEN-D-25-00399R1

Dear Mrs Beutel,

We are pleased to inform you that your manuscript 'Feasibility of a conversation-based brief intervention in general practice to reduce post-traumatic symptoms after intensive care treatment - a qualitative analysis of the PICTURE study.' has been provisionally accepted for publication in PLOS Mental Health.

Best regards,

Karli Montague-Cardoso

Staff Editor

PLOS Mental Health